

# Method for the quantitative evaluation of ecosystem services in coastal regions

Tomonari Okada[1], Yugo Mito[2], Erina Iseri[1], Toshiyuki Takahashi[2], Takanori Sugano[2], Yoshihiro B. Akiyama[1], Kenta Watanabe[3], Toko Tanaya[3], Hiroaki Sugino[4], Kanae Tokunaga[5], Takahiro Kubo[6] and Tomohiro Kuwae[3]

[1] Marine Environment Division/Coastal, Marine and Disaster Prevention Department, National Institute for Land and Infrastructure Management, Yokosuka, Japan
[2] Fukken Co., Ltd., Hiroshima, Japan
[3] Coastal and Estuarine Environment Research Group, Port and Airport Research Institute, Yokosuka, Japan
[4] Graduate School of Agricultural and Life Sciences, The University of Tokyo, Tokyo, Japan
[5] Ocean Alliance, The University of Tokyo, Tokyo, Japan
[6] Center for Environmental Biology and Ecosystem Studies, National Institute for Environmental Studies, Tsukuba, Japan

Corresponding author
Tomonari Okada,
okada-t92y2@mlit.go.jp

## ABSTRACT

Wetlands, tidal flats, seaweed beds, and coral reefs are valuable not only as habitats for many species, but also as places where people interact with the sea. Unfortunately, these areas have declined in recent years, so environmental improvement projects to conserve and restore them are being carried out across the world. In this study, we propose a method for quantifying ecosystem services, that is, useful for the proper maintenance and management of artificial tidal flats, a type of environmental improvement project. With this method, a conceptual model of the relationship between each service and related environmental factors in natural and social systems was created, and the relationships between services and environmental factors were clarified. The state of the environmental factors affecting each service was quantified, and the state of those factors was reflected in the evaluation value of the service. As a result, the method can identify which environmental factors need to be improved and if the goal is to increase the value of the targeted tidal flat. The method demonstrates an effective approach in environmental conservation for the restoration and preservation of coastal areas.

Subjects Conservation Biology, Ecology, Ecosystem Science, Science Policy, Coupled Natural and Human Systems
Keywords Ecosystem services, Quantitative evaluation, Wetland restoration, Ecosystem-based management, Sustainability, Tidal flat, Wetland, Artificial tidal flat

## INTRODUCTION

Evaluating ecosystem services highlights the complexity of the relationship between society and ecosystems and clarifies how human decisions can affect the value of ecosystem services (*Millennium Ecosystem Assessment (MEA), 2005*). Such evaluations are needed to express any change in value in units (e.g., monetary) that enable the value of ecosystem services to be incorporated into public decision-making processes

(*Daily, 1997*; *Mooney, Cropper & Reid, 2005*). Because many ecosystem services are mixed with public goods, regulating their use is difficult even when they are nearly or completely exhausted (*De Groot et al., 2010*). In addition, because many ecosystem services are produced and enjoyed in the absence of market transactions, their value is often underestimated and even ignored in daily decision-making (*Pascual et al., 2010*). Furthermore, because many people enjoy the benefits of ecosystem services without even being aware of them, they are often unable to recognize the ecosystem services' value and importance.

For these reasons, demonstrating the values of ecosystems is required and monetization of ecosystem services has been proposed (*Costanza et al., 1997*; *Daily, 1997*). Conventional methods to do so include direct market valuation approaches, revealed preference approaches, and stated preference approaches (*Farber, Costanza & Wilson, 2002*; *Pascual et al., 2010*). Applying direct market valuation and revealed preference approaches are expected to provide reasonable evaluations of services for which sufficiently developed markets already exist, but not for services for which no or only nascent markets exist, in which case the use of stated preference approaches is preferred (*Chan, Satterfield & Goldstein, 2012*; *Cooper et al., 2016*).

*Halpern et al. (2012)* proposed the Ocean Health Index (OHI) as a method for comprehensively quantitating ocean health. The OHI defines healthy oceans as those that sustainably deliver a range of benefits to people now and in the future. The index enables the comprehensive evaluation of not only benefits gained from ecosystems but also social and economic benefits (*Halpern et al., 2014*). The OHI scores the state of each ecosystem service against its reference point (*Samhouri et al., 2012*), which enables the degree of achievement of each service to be evaluated. The OHI also enables sustainability and changes in policy or regulatory systems to be considered, and it can be adapted flexibly to match the quantity and quality of data, as well as changes in geographic scale (*Elfes et al., 2014*; *Halpern et al., 2015*; *Lowndes et al., 2015*; *Selig et al., 2015*). One key advantage of the OHI over other approaches is that it enables the scoring and quantification of services that are difficult to quantify in economic terms.

Biodiversity is the foundation for ecosystem services, and the loss of biodiversity causes the value of ecosystem services and productivity to decline (*Worm et al., 2006*), which is a global issue. In 1992, the Convention on Biological Diversity was adopted as an international framework for comprehensively preserving biodiversity and promoting the sustainable use of biological resources. Efforts are being made in coastal regions around the world to create artificial wetlands, tidal flats, seaweed beds, and coral reefs with the ultimate goal of restoring and preserving coastal habitats (*PIANC, 2008*; *Fredette & Suedel, 2011*; *De Vriend & Van Koningsveld, 2012*; *International Association of Dredging Companies (IADC), 2017*). In addition, efforts are also being made to create habitats in harbors by adding habitat functions to port structures such as breakwaters and seawalls (*Kamimura et al., 2011*).

Wetlands, tidal flats, seaweed beds, and coral reefs are valuable places not only as habitat for many species, but also as places where people interact with the sea and access many ecosystem services. Environmental improvement projects for these areas should therefore be evaluated from the perspectives of habitat and ecosystem services.

However, these kinds of environmental improvement projects are relatively small in scale when considering ecosystem services, and past evaluation methods of ecosystem services are conducted on large spatial scales (e.g., global, national, regional) and are not suitable for the evaluation of these kinds of environmental improvement projects. In addition, because the evaluation of ecosystem services by environmental economic methods is not linked to either the natural systems or the related social systems of wetlands, tidal flats, seaweed beds, and coral reefs, the need for countermeasures to improve the areas cannot be demonstrated. Moreover, because ecosystems change over time after the creation of an artificial tidal flat or wetland (*Palmer, Ambrose & Poff, 1997*; *Kuwae, 2005*), the effect of the environmental improvement project must be evaluated for several years following completion. Because the effect of the environmental improvement project strongly depends on ecosystem conditions, it is necessary to carefully consider those conditions to conduct an appropriate evaluation of a project (*Zhao et al., 2016*). For these reasons, evaluation of environmental improvement projects requires a method, that is, capable of considering both the relevant natural and social systems and incorporating the concept of sustainability.

In response to these problems, we are working on a two-step method in which (1) ecosystem services are scored reliably and (2) a composite evaluation is conducted, taking the weighted service scores into consideration. Including various kinds of environmental improvement projects in the early stages of method development would complicate the process, and artificial tidal flats are being constructed in many coastal regions. Therefore, we initially focused only on tidal flats and quantified the services provided by tidal flats and tidal flat ecosystems.

The specific aim of this study, as a first step toward the larger goal, was to develop an evaluation method coastal ecosystem services index (CEI) that can quantify services and sustainability trends as well as identify relevant environmental factors for each service and those in need of countermeasures by fully considering both the natural systems and social systems in the surrounding area (i.e., waters) of an environmental improvement project.

## METHODS

### Evaluated tidal flats and evaluation year

In this study, we evaluated two artificial tidal flats and two natural tidal flats in Tokyo Bay, an enclosed bay: Shiosai Nagisa (SN) and Umi Koen (UK), the artificial tidal flats, and Tama River tidal flat (TR) and Obitsu River tidal flat (OR), the natural tidal flats (Figs. 1 and 2; Table 1). With this method, the value of a tidal flat is scored by comparison with a reference point (see section 'Calculation method for the service score'). Therefore, when evaluating an artificial tidal flat, it is essential to also evaluate the state of a natural tidal flat to be used as the point of reference. It is also necessary to set the target and reference tidal flats within the same overall water area. Although four tidal flats were targeted in this study, it would generally be desirable to use more target tidal flats to reduce the deviation of the service scores.

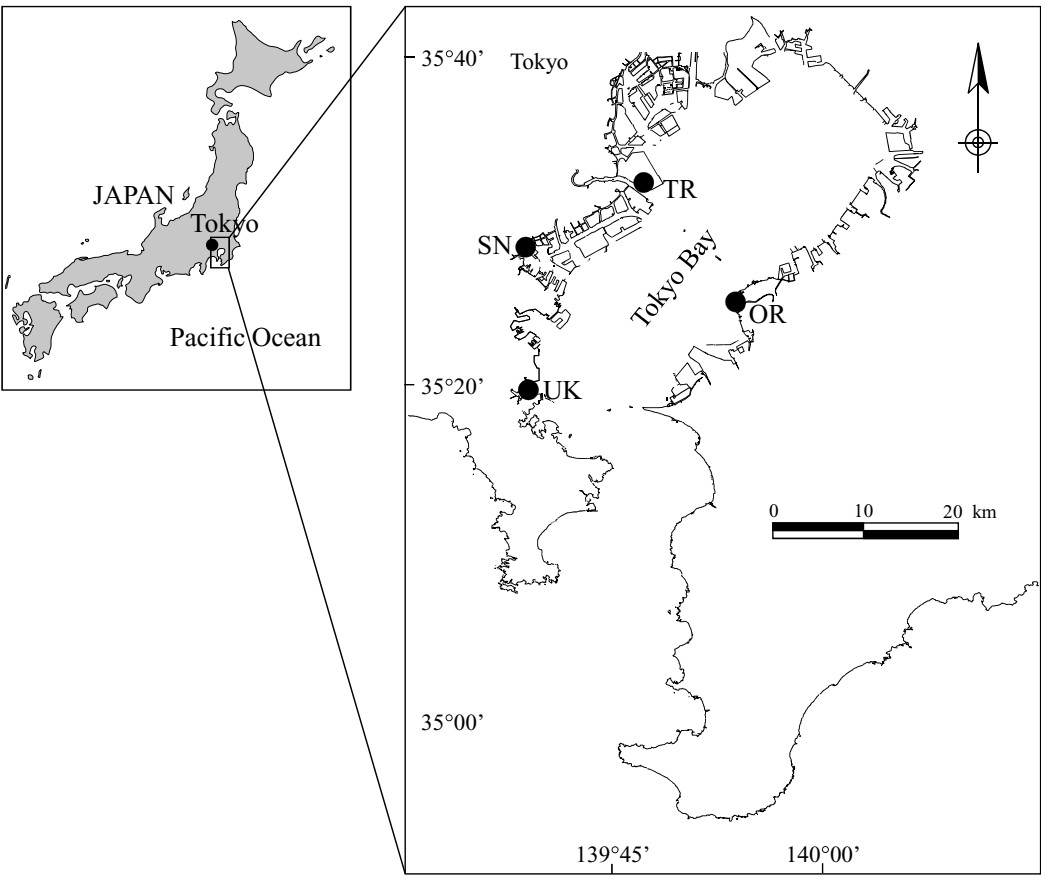

**Figure 1 Location of the four tidal flats in Tokyo Bay: UK (Umi Koen), SN (Shiosai Nagisa), TR (Tama River tidal flat), and OR (Obitsu River tidal flat).**

Shiosai Nagisa is an artificial biological symbiotic port structure in Tokyo Bay (Figs. 1 and 2A) (*Morohoshi et al., 2008*). To create habitats near the seawall, which has an upright surface, a terrace-type tidal flat was placed in front of the seawall. To create biodiversity in the depth direction, the ground height of the tidal flat was set at three heights (DL = 0, +0.5, +1.0 m). In addition, this terrace-type tidal flat also serves a disaster-prevention function for the seawall. SN is a demonstration facility, but it is not usually open to the public and access is possible only at events such as environmental education. UK is an artificial tidal flat (Figs. 1 and 2B) that was completed more than 30 years ago. It can be expected that its functions and features are intermediate between natural and artificial tidal flats. It is familiar to residents because it has a public-access beach in a crowded urban area. The other two study areas are natural tidal flats in the bay (Figs. 1, 2C and 2D).

The scope of the tidal flat evaluation comprised the area from the water–land interface to the intertidal zone (i.e., the area shallower than the low water level). The water–land interface was delineated by embankments or structures abutting the landward side of tidal flats.

The evaluation year was 2013, with data from 2009 to 2012 being used as past values to calculate the trend scores.
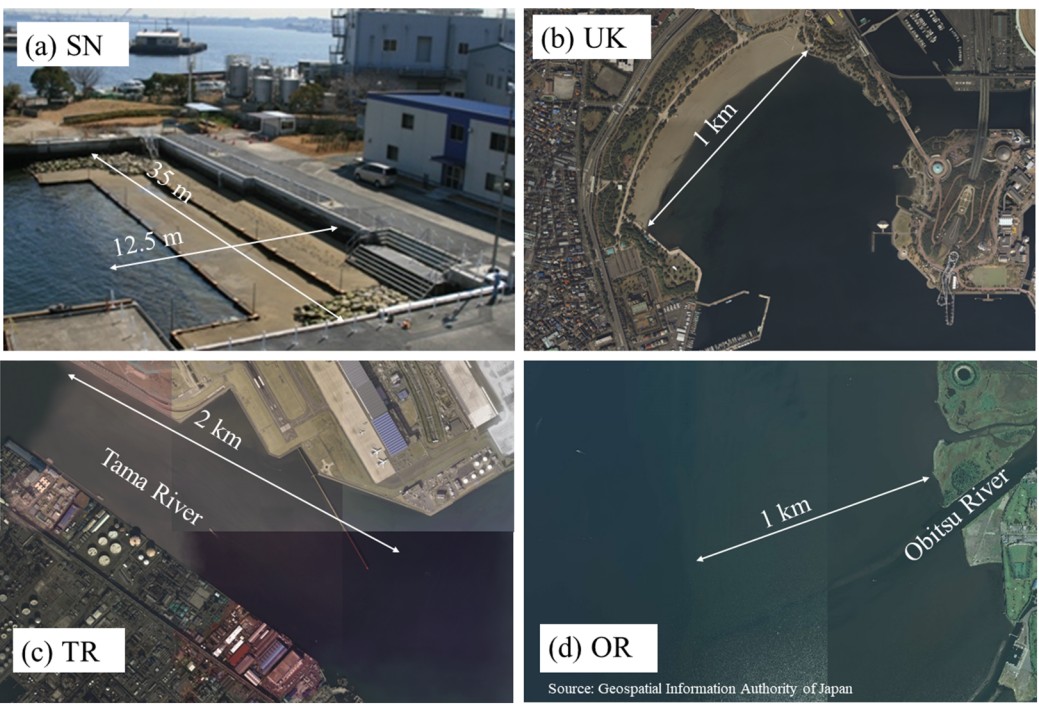

**Figure 2** **Photos of the four tidal flats: (A) SN, (B) UK, (C) TR, and (D) OR.** (D) source: aerial photograph of Geospatial Information Authority of Japan, https://mapps.gsi.go.jp/.

**Table 1** **Characteristics of each evaluated tidal flat in this study.**

|  | SN (Shiosai Nagisa) | UK (Umi Koen) | TR (Tama River tidal flat) | OR (Obitsu River tidal flat) |
|---|---|---|---|---|
| Natural or artificial | Artificial (completed in 2008) | Artificial (completed in 1980) | Natural | Natural |
| Sediment condition | Sand | Sand | Sand and mud | Sand and mud |
| Area (m²) | 438 | 150,000 | 250,000 | 6,500,000 |
| Coastline (m) | 35 | 800 | 2,500 | 3,500 |

## Setting of services

*Halpern et al. (2012)* constructed the OHI, which covered 10 goals encompassing 13 sub-goals from the standpoint of site use: food provision (fishing, mariculture); artisanal fishing opportunities; natural products; carbon storage; coastal protection; coastal livelihoods and economies (jobs, wages, economy); tourism and recreation; sense of place (iconic species, lasting special places); clean waters; and biodiversity (habitats, species). Taking the characteristics of tidal flats into consideration, we constructed an index based on six services (12 sub-services): food provision; coastal protection; water front use (recreation, environmental education, research); sense of place (historical designation as special sites, place for everyday rest and relaxation); water quality regulation (removal of suspended matter, organic matter decomposition, carbon storage); and biodiversity (degree of diversity, rare species) (Table 2).
**Table 2 Services, indices of services provided, definitions of spatial and temporal range, and key index unit of tidal flats and tidal flat ecosystems.**

| Service (sub-service) | | Index | Definition of spatial and temporal range | Key index unit |
|---|---|---|---|---|
| Food provision | | Wet weight of commercially important species | Annual average per unit area | $g/m^2$ |
| Coastal protection | | Wave energy reduction ratio, coast to offshore | Annual average in evaluation area | $N/m$ |
| | | Wave run-up height ratio of run-up height to top of seawall | Annual maximum in evaluation area | $m$ |
| Water front use | Recreation | Number of visitors for the purpose of recreation | Annual total number per unit area | $people/m^2/year$ |
| | Environmental education | Number of visitors for the purposes of environmental education | Annual total number in evaluation area | $people/area/year$ |
| | Research | Number of papers and reports | Annual total number in evaluation area | $papers/area/year$ |
| Sense of place | Historical designation as special sites | Number of rites and festivals and faith-related buildings | Annual total number in evaluation area | $times/area/year$ |
| | Places for everyday rest and relaxation | Total duration of stay for everyday use adjusted by an awareness factor for the value of the tidal flat | Annual total number in evaluation area | $hour/area/day$ |
| Water quality regulation | Suspended material removal | Bivalve water filtration volume | Annual average per unit area | $g/m^2/day$ |
| | Organic matter decomposition | COD purification amount by benthic organisms | Annual average per unit area | $g\text{-}COD/m^2/day$ |
| | Carbon storage | Carbon fixation in benthic organisms and sediment | Annual minimum per unit area | $g\text{-}C/m^2$ |
| Biological diversity | Degree of diversity | Diversity Index (H') | Annual average in evaluation area | – |
| | Rare species | Number of threatened species adjusted by endangered category type | Annual total number in evaluation area | $number\ of\ species/area/year$ |

**Note:**
The services and indices are described in more detail in the Supplemental Information.

## Calculation method for the service score

To quantify services, we applied the calculation process similar to the one used in OHI. A service score $I_i$ for each service $i$ is calculated from the present status $x_i$ and the likely near-term future status $x_{i,F}$ with Eq. (1):

$$I_i = \frac{x_i + x_{i,F}}{2}, \tag{1}$$

The present status $x_i$ is normalized by the reference point with Eq. (2):

$$x_i = \frac{X_i}{X_{i,R}}, \tag{2}$$

where $X_i$ is the present status value for service $i$ and $X_{i,R}$ is the reference point. Any $X_i$ value beyond $2\sigma$ from the mean was determined to be an outlying observation and was not used in the calculations. *Halpern et al. (2012)* proposed four methods for setting OHI reference points that used: (1) a known functional relationship, (2) a time series approach, (3) a spatial comparison, and (4) an established target value or standard. Because it was difficult to generalize reasonable reference points such as those generated by methods (1) and (4) for specific habitats where conflicts between stakeholders more specifically occur, we used the maximum observed value for each index in the four tidal flats in the most recent 5-year period as the reference point for that index. As described below, this timeframe matches that used to calculate the likely status in the near-term future.

Setting reference points to the "best" conditions, as is done in the OHI, may result in an evaluation result, that is, easy to interpret, but if the reference points are not agreed upon by all stakeholders, it could lead to erroneous information being given to decision-makers. For this reason, CEI does not use reference points based on social consensus; they are set as noted previously and then used as a relative evaluation criterion. When evaluating a tidal flat service, rather than aiming at a "best" reference point, it is important to evaluate the level of the target tidal flat service compared with that of other tidal flats. With this procedure, it is possible to evaluate the status of the services of the target tidal flats even if the reference point does not represent a "best" condition. Using this type of relative method of setting reference points seems to be the most appropriate in this study, especially because we are comparing artificial and natural tidal flats. In addition, for the purpose of evaluating the status of the artificial tidal flats, it is important to compare them with the status of natural tidal flats in the current ecological environment. Therefore, a period of 5 years was considered to be appropriate in the calculations. On the other hand, if the purpose was to evaluate the status of natural tidal flats in waters where the environment had deteriorated 50 years ago (e.g., in Tokyo Bay), a relative comparison using a maximum value in the past 5 years would not be appropriate. In that case, an ideal value before the environment had deteriorated (i.e., more than 50 years ago) should be used as a reference point.

The likely near-term future status $x_{i,F}$ for service $i$ is defined by Eq. (3):

$$x_{i,F} = (1 + \beta T_i + (1 - \beta)PR_i) \cdot x_i, \tag{3}$$

where $T_i$ is the trend for service $i$ (see Section Calculation of the trend), $\beta$ is the relative importance of the trend vs pressure and resilience (PR) scores, and $PR_i$ is the PR score for service $i$ (see Section Development of conceptual model and calculation of the PR scores). In this study, we used $\beta = 0.67$ (*Halpern et al., 2012*), weighting the directly measured past trend by a factor of 2:1 over the indirectly measured PR scores. However, because our method emphasizes the link between environmental factors and scores, we plan to reconsider the value of $\beta$ in future work.

## Calculation of the trend

The trend $T_i$ ranges from $-1.0$ to $+1.0$ and indicates whether the measured values for each service over the most recent 5 years show an upward or downward trend. The trend is

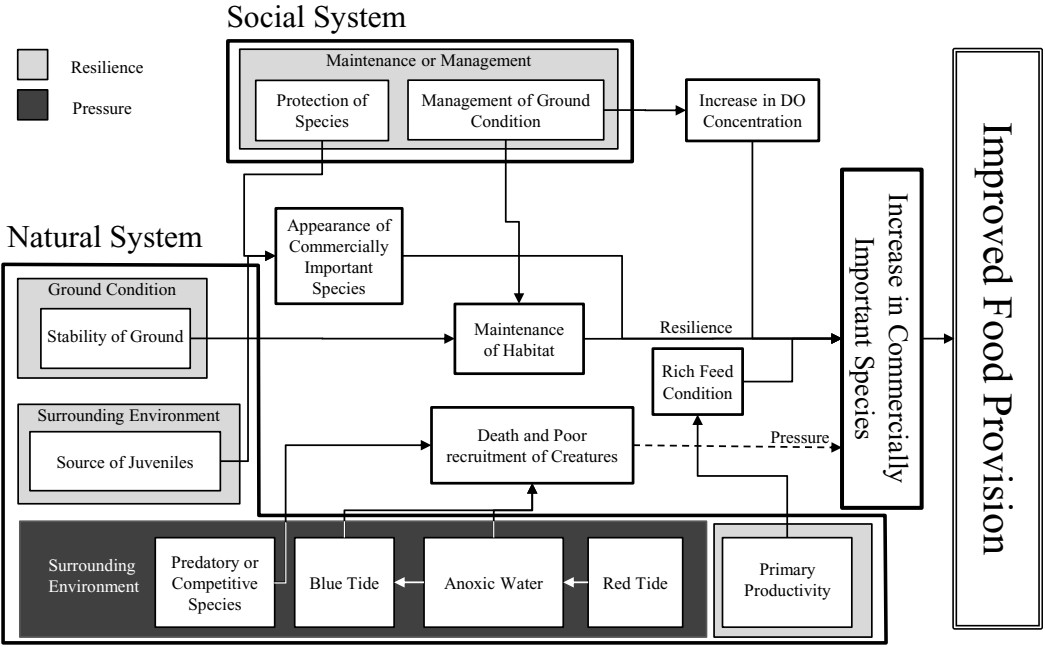

**Figure 3 Conceptual model of environmental factors for food provision.**

derived from the slope of the straight line fit to the relevant time series data. When the trend could not be calculated owing to the absence of data for the past 5 years, it was set to 0.

## Sustainability score

The difference between the present status $x_i$ and likely near-term future status $x_{i,F}$ represents the increase or decrease of the service in the next 5 years; this is defined as the sustainability score $S_i$:

$$S_i = \frac{\left(x_{i,F} - x_i\right)}{x_i} \tag{4}$$

A positive sustainability score means that the service will improve under present conditions, and a negative one means that the service will decline under present conditions. We can look for the environmental factors underlying negative sustainability scores in the PR scores and appropriate countermeasures can be taken for the negative environmental factors.

## Development of conceptual model and calculation of the pressure and resilience scores

To obtain useful information for the management of tidal flats, we developed a conceptual model of the natural and social systems considering environmental factors related to a service (Fig. 3; Table 3). This conceptual model is very important in guiding the use of appropriate countermeasures. In addition, environmental factors must be properly selected depending on the target waters.

| Table 3 Environmental factors for food provision. | | |
|---|---|---|
| **Environmental factor** | | **Pressure or resilience condition** |
| Anoxic water | Resilience | DO concentration >4 mg/L |
| | Pressure | DO concentration <4 mg/L |
| Blue tide | Resilience | No occurrence of blue tide |
| | Pressure | Occurrence of blue tide |
| Predatory or competitive species | Resilience | No predatory or competitive species against commercially important species |
| | Pressure | Existence of predatory or competitive species against commercially important species |
| Primary productivity | Resilience | Chl-a concentration >3 µg/L |
| | Pressure | Chl-a concentration <3 µg/L |
| Stability of ground | Resilience | Stable ground |
| | Pressure | Low stability of the ground (erosion, deposition, consolidation, subsidence, etc.) |
| Source of juveniles | Resilience | Existence of a nearby tidal flat |
| | Pressure | No nearby tidal flat |
| Management of ground condition | Resilience | Existence of management of ground condition |
| | Pressure | No management of ground condition (e.g., sand fill, flowing) |
| Protection of species | Resilience | Existence of efforts to protect species |
| | Pressure | No efforts to protect species |

Here, environmental factors can have a positive effect (enhance resilience) or negative effect (increase pressure) depending on their magnitude or presence/absence. This is where our definitions of PR differ from those of *Halpern et al. (2012)*. Using food provision as an example, *Halpern et al. (2012)* generated separate PR scores based on factors such as fishing and habitat destruction for the former and marine protected and fishing resource management areas for the latter. In this study, we combined PR into a single PR score where $PR_{i,j}$ takes a value between −1.0 and 0.0 when the overall effect is negative and between 0.0 and +1.0 when the overall effect is positive; $j$ indicates environmental factor $j$. In addition, the method for evaluating $PR_{i,j}$ differs depending on whether the data being used are quantitative or qualitative.

In the case of environmental factors such as dissolved oxygen (DO) concentration and chlorophyll a (Chl-a) concentration for which measured values are available, we set threshold values based on previous research and criteria and calculated $PR_{i,j}$ with evaluation models that would yield a $PR_{i,j}$ of −1.0 when the environmental factor was unfavorable and +1.0 when the environmental factor was favorable.

In the case of DO concentration, using the *Ministry of the Environment (2016)* criterion as a basis, a $PR_{i,j}$ of −1.0 was assigned when the DO concentration was ≤2 mg/L and a $PR_{i,j}$ of +1.0 when the DO concentration ≥6 mg/L. Intermediate values (between −1.0 and +1.0) were assigned for DO concentrations between two and six mg/L (Fig. 4). Accordingly, DO concentrations <4 mg/L yielded negative $PR_{i,j}$ and were
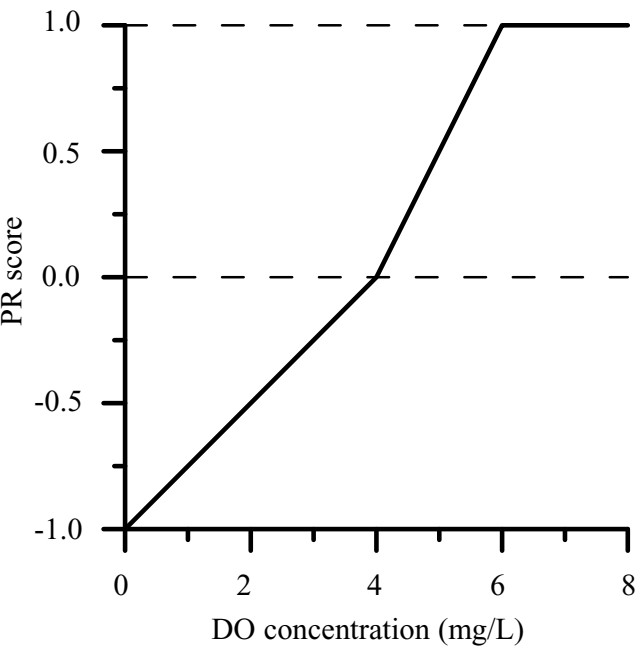

**Figure 4 PR score evaluation model for DO concentration.**

considered pressures, whereas DO concentrations >4 mg/L yielded positive $PR_{i,j}$ and were considered resilience factors.

Other environmental factors such as ground stability and blue tide (*Furukawa & Okada, 2006*) were evaluated from qualitative conditions because it is difficult to quantitatively indicate a degree for these factors. $PR_{i,j}$ was assigned a score of +0.5 if the positive factor occurred (or existed) and a score of −0.5 if the negative factor occurred (or existed). Because the impacts of such environmental factors vary in magnitude, it would be ideal to assign scores ranging between −1.0 and +1.0 depending on the degree of impact. However, there are insufficient data and knowledge to be able to quantitatively differentiate impacts. Accordingly, in this study, the influence of uncertain environmental factors on the scores was assumed to be half that of certain environmental factors, and we assigned either −0.5 or +0.5 for these factors.

$PR_i$ represents the average of all $PR_{i,j}$ scores impacting service $i$. Although the weights of each environmental factor were assumed to be the same in this study, we plan to study the weighting of the influence of environmental factors on services in the future.

## Setting of indices

For each service, we identified indices that accurately represent the state of the service being assessed and for which data are available (Table 2).

As an index of food provision, we used the wet weight of commercially important species to evaluate the supply capacity of commercially important species. Details are shown in the Supplemental Information 1. As indicators of coastal protection, we used the wave energy reduction ratio and the wave run-up height ratio to evaluate an area's flood

prevention function through the reduction of wave surges and wave overtopping (Supplemental Information 2).

As an index of recreation for water front use, we used the number of visitors for the purpose of recreation and evaluated the use of tidal flats for shellfish gathering and other activities (Supplemental Information 3). As an index of environmental education for water front use, we used the number of visitors for the purpose of environmental education and related activities (Supplemental Information 4). As an index of research for water front use, we used the number of published papers and reports (Supplemental Information 5).

As an index of historical designation as special sites for sense of place, we used the numbers of festivals and of faith-related buildings (Supplemental Information 6). As an index of places for everyday rest and relaxation for sense of place, we developed a rest and relaxation index relative to the total hours of everyday use that was adjusted for the user's stated level of conscious awareness of the value of the sites for walks, rest and relaxation, and other similar uses (Supplemental Information 7).

As an index of suspended material removal for water quality regulation, we used the bivalve water filtration volume (Supplemental Information 8). As an index of organic matter decomposition for water quality regulation, we used the chemical oxygen demand (COD) purification amount (calculated from the production/biomass ratio) by benthic organisms (Supplemental Information 9). As an index of carbon storage for water quality regulation, we used the carbon fixation in benthic organisms and sediment (0–10 cm in depth) (Supplemental Information 10).

As an index of degree of diversity for biological diversity, we used the Shannon–Wiener diversity index (H') for the entire study area (Supplemental Information 11). Finally, as an index of rare species for biological diversity, we used the number of threatened species adjusted by category of threatened status (Supplemental Information 12).

## Spatial scale and time scale

There are two approaches to the spatial scale in calculating services: evaluate either the amount of service obtained from the entire tidal flat or the amount of service obtained from a unit area. The first type of evaluation result is obviously highly dependent on the area of the tidal flat. The second type does not demonstrate the total amount of service, but it does show the quality of the tidal flat.

For these reasons, CEI basically uses values per unit area. However, indices with weak dependency on area and those that are difficult to correct for phenomena per unit area were based on the amount of service obtained from the entire tidal flat (Table 2). The number of species has a nonlinear relationship with area (*Gotelli & Colwell, 2001*), and it is difficult to formulate a relation. Therefore, with regard to the number of species, the survey results were used without spatial correction. When evaluating biodiversity, we used data on the number of species obtained by the same survey method to reduce bias caused by a difference in the number of survey sites between tidal flats as much as possible. Introducing an appropriate method for spatial correction of the number of species remains to be dealt with in the future.
The time scale was set to 1 year. However, because the time correction method depends on the characteristics of the index, it differs for each indicator (Table 2). An annual average value was used as the index value for an index where the annual average value is important, such as the existing number of clams or amount of carbon storage. For an index where annual total is important, such as the number of visitors and rare species, the annual total value was used as the index value. For an index where the annual maximum value is important, such as the relative wave run-up height for coastal protection, the annual maximum value was used as the index value.

## RESULTS

The service scores are summarized in Table 4. Details of the calculation process and more detailed results for each area are presented in the Supplemental Information. Because the same reference point was used for all four tidal flats when quantifying progress toward each service, it is possible to compare service scores across tidal flats. In addition, for each service, areas in need of countermeasures and where they could prove effective can be identified from the sustainability scores and PR scores of the environmental factors.

### Food provision

The food provision service score of SN (12.4) was higher than those of the natural tidal flats (TR, 4.0, and OR, 5.6), indicating a higher quality (wet weight per unit area) (Table 4 and Supplemental Information 1). Although SN had a high service score, its sustainability score was negative (−41%), indicating that this service will decay under the present condition. To suppress this decay, countermeasures need to be taken in the categories of anoxic water, blue tide, ground stability, predatory or competitive species, and protection of species, all of which had negative PR scores (Fig. S3). Although the service score of UK was low (5.3), its sustainability score was positive (+17%), so the present status can be maintained in the present environmental condition. UK is located in an area with good water quality and has been established for more than 30 years, so there is no need for countermeasures against the water environment and instability of the ground just after construction. The service scores of TR and OR (4.0 and 5.6) were not high, but the sustainability scores were positive (+1% and +12%) (Table 4), so their present status can be maintained in the present environmental condition. To improve the service of TR, attention could be paid to predatory or competitive species (Fig. S3), and in OR, anoxic waters should be a point of attention.

### Coastal protection

Coastal protection was evaluated from two indicators, the maintenance effect of tidal flats owing to wave energy reduction and the suppression effect of wave run-up height by a seawall behind tidal flats (Supplemental Information 2). The maintenance effect of tidal flats owing to wave attenuation was strong in large tidal flats, but the difference between the artificial tidal flats was small because the artificial tidal flats were basically arranged in waters with small waves (Table S5). There was little difference in the

**Table 4 Calculated service and sustainability scores for all services.**

| Service (sub-service) | | Service score ($I_i$)/Sustainability score ($S_i$) | | | |
|---|---|---|---|---|---|
| | | SN | UK | TR | OR |
| Food provision | | 12.4/−41% | 5.3/+17% | 4.0/+1% | 5.6/+12% |
| Coastal protection | | 81.2/0% | 96.4/+13% | 85.1/−4% | 94.3/+4% |
| Water use | Recreation | −/− | 78.7/−1% | 0.3/−7% | 0.3/+16% |
| | Environmental education | 40.3/−41% | 66.8/+11% | 0.0/− | 39.4/+17% |
| | Research | 64.7/+26% | 14.5/+2% | 75.1/+34% | 70.8/+48% |
| Sense of place | Historical designation as special sites | −/− | −/− | 50.8/+3% | 80.0/+13% |
| | Places for everyday rest and relaxation | −/− | 100/+10% | 3.2/−3% | 2.8/+17% |
| Water quality regulation | Suspended material removal | 100/+18% | 20.7/+13% | 14.3/+9% | 8.9/+10% |
| | Organic matter decomposition | 52.1/−8% | 12.0/+27% | 40.2/+22% | 17.8/+19% |
| | Carbon storage | 58.7/−9% | 16.6/−3% | 79.7/−8% | 13.6/+3% |
| Biological diversity | Degree of diversity | 86.5/+10% | 100/+28% | 87.9/−2% | 78.1/+2% |
| | Rare species | 7.8/−14% | 12.0/+16% | 100/+54% | 100/+31% |

**Notes:**
Gray shading highlights a negative sustainable index score.
A "−" indicates the tidal flat was omitted from the analysis, usually because the service did not apply in that tidal flat.

suppression service score, most likely because it had been thoroughly investigated at the design stage of the seawall (Table S6). In other words, the artificial tidal flat and the seawall are designed to satisfy the function of coastal protection. From this point of view, we consider the sustainability score to be more important in this category than the service score (Table 4). The sustainability scores of SN and UK, the artificial tidal flats, were not negative (0%, +13%), and countermeasures are therefore considered to be unnecessary. The sustainability score of TR was negative (−4%), mainly because management of ground condition and inspection and repair of seawall have not been implemented (Fig. S7).

## Recreation (water front use)

Because SN is within a demonstration area, that is, not open to the public, recreation is not allowed and, therefore, there is no service score for this category. Because UK is used by many people, its service score was extremely high (78.7), but the service scores of the other tidal flats were less than 1 point (Table 4; Supplemental Information 3). However, deterioration in the habitat (such as a massive growth of sea lettuce, *Ulva* sp.) was observed (Fig. S9), so there is concern that the sustainability score was negative (−1%). The sustainability score of TR was also negative (−4%). To maintain or enhance the present status of Recreation in the natural tidal flats, the factors management group, attracting visitors, publicity work, incidental facilities, and accessibility need attention (Fig. S9). However, for natural tidal flats, stakeholders need to be consulted when deciding whether promotion of recreation is appropriate because there may be trade-offs involved. The trade-off relationships between these services should be examined in future research.

### Environmental education (water front use)

The service scores of environmental education for SN and UK (40.3 and 66.8) were higher than those of the natural tidal flats (Table 4; Supplemental Information 4). However, the sustainability score of SN was negative (−41%). The factors attracting visitors, publicity work, accessibility, protection of species, healthy habitat, and stability of ground were indicated as needing improvement to maintain the service of environmental education (Fig. S11). At TR, no environmental education was carried out and the service score was zero, in large part because there is no organization such as a non-profit organization to implement a relevant program. The service score of OR (39.4) was similar to that of SN, and its sustainability score was also high (+17%), indicating that environmental education is active in this area and the condition of the surrounding environment is also good.

### Research (water front use)

The natural tidal flats, TR and OR, had higher service scores (75.1 and 70.8) for research than the artificial tidal flats (Table 4; Supplemental Information 5). As natural scientific research subjects, natural tidal flats may be more attractive. At the demonstration facility SN, regular testing and surveillance were conducted, and the service score was close to that of the natural tidal flats (64.7). Because research is established with various interests and purposes, its PR index was not evaluated. For this reason, the sustainability scores only show only recent trends and have little meaning when considering countermeasures.

### Historical designation as special sites (sense of place)

Because this service is unique to the given area, it does not exist in artificial tidal flats, which have no cultural history and have usage constraints as public facilities (Table 4; Supplemental Information 6). However, it is possible that this service may exist in an artificial tidal flat that was created as a mitigation of a natural tidal flat that had the service.

### Places for everyday rest and relaxation (sense of place)

Because SN is within an experimental area, that is, not open to public, this service is restricted, similar to recreation (Table 4; Supplemental Information 7). On the other hand, UK is used by many people and its service score was extremely high (100).
The sustainability score was also positive (+10%), so the area can continue to provide a high level of service under present environmental conditions. Among the PR indexes, only the healthy habitat factor was negative (Fig. S19). By implementing countermeasures related to this factor, it could be possible to provide an even higher level of service.
The service scores of TR and OR, the natural tidal flats, were low (3.2 and 2.8).
The sustainability score of TR was negative (−3%), and the factors management of ground condition, management groups, and incidental facilities were indicated as needing countermeasures. The OR sustainability score was positive (+17%), so the present status can be maintained under present environmental conditions. The low service score was likely influenced by the small number of people living in the surrounding area.

### Suspended material removal (water quality)

The SN service score was high (100) for material removal (Table 4; Supplemental Information 8). In addition, its sustainability score was positive (+18%). Although the present environmental condition is sufficient, countermeasures against anoxic water, blue tide, and stability of ground would be effective to enhance the service (Fig. S22). The UK service score (20.7) was not as high that of SN, but its sustainability score (+13%) and PR index were high and it is expected to maintain a stable service. The service scores of both natural tidal flats were not as high (14.3 and 8.9 for TR and OR, respectively), but both had positive sustainability scores (+9% and +10%), so a stable level of service can be expected.

### Organic matter decomposition (water quality)

Although this SN service score was high (52.1), the sustainability score was negative (−8%) (Table 4; Supplemental Information 9). To maintain this service, countermeasures for anoxic water, blue tide, and stability of ground should be considered (Fig. S24). In contrast, the UK service score was low (20.7), but the sustainability score was positive (+13%). The TR service score (40.2) was high and its sustainability score was positive (+22%), so a stable level of service can be expected. Although the service score of OR (17.8) was not as high as that of TR, the sustainability score was positive (+19%), so stable service can be expected to be maintained.

### Carbon storage (water quality)

The service score of SN (58.7) was high, but the sustainability score was negative (−9%) (Table 4; Supplemental Information 10) because the trend index had a large negative value (Table S36). To enhance the carbon storage service, it is important to increase the amount of carbon stored in the bottom sediment, which is responsible for about 90% of the carbon storage function, and it is necessary to take countermeasures related to ground stability (Fig. S27). The service score of UK (16.6) was low and its sustainability score was negative (−3). To enhance its carbon storage service, the accumulation of organic matter in sediment by sand cupping should be promoted. The service score of TR (79.7) was high, but its sustainability score was negative (−8%). Although the service score of OR (13.6) was low, its sustainability score was positive (+3%), indicating it is expected to maintain a low but stable level of service.

### Degree of diversity (biodiversity)

The service scores of both SN and UK (86.5 and 100) were equal to or higher than those of the natural tidal flats, and the sustainability scores of both were positive (+10% and +28%) (Table 4; Supplemental Information 11). The PR index of SN was negative (Table S39), however, and it would be desirable to pay attention to the factors alien species, surrounding environment, stability of ground, and protection of species (Fig. S29). The service scores of TR and OR (87.9 and 78.1) were also high, but the sustainability score of TR was negative (−2%). To maintain this high level of service, it is necessary to pay particular attention to alien species and protection of species.

### Rare species (biodiversity)

The service scores of SN and UK (7.8 and 12.0) were lower than those of TR and OR (Table 4; Supplemental Information 12). It is probably inevitable that the rare species service would be lower in artificial tidal flats as compared with natural tidal flats. However, to at least approach the service level of natural tidal flats, important factors are predatory or competitive species, alien species, healthy habitat, stability of ground, and protection of species in SN and alien species and healthy habitat in UK (Fig. S31).

## DISCUSSION

### Characteristics and features of the method

By comparing the service scores among multiple tidal flats, we were able to evaluate the service of each tidal flat. By combining the service scores with a sustainability score, it was possible to evaluate whether the service was sustainable or deteriorating under present environmental conditions. If the level of service was low or there was concern about service deterioration, we could use the PR scores to identify the relevant environmental factor(s) for which it would be desirable to implement countermeasures. Of course, whether this environmental factor is valid or not depends on whether the conceptual model has been properly designed and implemented. At a later practical stage, it will be necessary to review the conceptual model. Incidentally, the conceptual model shown in this study is a model for tidal flats in Tokyo Bay, which has undergone remarkable eutrophication. It is important to create a conceptual model, that is, appropriately based on the characteristics of the applicable water area.

For the artificial tidal flat SN, the sustainability scores of food provision, environmental education, organic matter decomposition, carbon storage, and rare species were negative, indicating these services needed countermeasures to become sustainable. Environmental factors identified as requiring countermeasures were anoxic water, blue tide, stability of ground, predatory or competitive species, protection of species, attracting visitors, publicity work, accessibility, healthy habitat, and alien species. Among these, ground stability was a common environmental factor identified for all services, and anoxic water and blue tide were common environmental factors in need of countermeasures for food provision, organic matter decomposition, and carbon storage. It is therefore clear that some environmental factors are applicable to multiple services.

At this stage, the relative "weight" of service has not been considered. In other words, the value of 100 points differs depending on the service, and it is impossible to compare values between services using only the values presented in Table 4. Also, it is not possible to simply integrate the service scores and give a total score within a tidal flat. Therefore, at this stage, it is unknown which countermeasures will effectively improve the overall level of service of the tidal flat. When considering the use of countermeasures, it is necessary to consider which ones will be most effective based on the contribution of relevant environmental factors to multiple services and a comprehensive evaluation of services.

Regarding such a comprehensive evaluation, Halpern et al. (2012) assumed that all services had the same weight, and the average score of all services was taken as a total score.

In addition, *Daigle et al. (2017)* calculated an overall score by setting service weights based on the importance of each service obtained from the results of a questionnaire administered to residents in the target evaluation area. In response to this problem, we are considering a weight-setting method based on economic value and questionnaire survey results.

As mentioned in Section Recreation (water front use), there may be a trade-off between services. Too little information was obtained from these four tidal flats to show any trade-off relationships. It is therefore necessary to clarify any trade-offs by increasing the number of study areas. Currently, our results allow consideration of countermeasures to improve individual services, but, that is not sufficient to improve the comprehensive evaluation of services of tidal flats. Incorporating trade-off relationships and a weighting of services is necessary to be able to consider which services would be most effective for taking countermeasures.

It is conceivable that the weight of the effect of the environmental factors also differs. At present, the PR scores were all weighted the same, but we need to consider weighting these scores as well. In addition, we assumed a qualitative PR score to be half that of a quantitative PR score. This assumption should be evaluated together with the weighting of scores within the PR index. Finally, the coefficient $\beta$, which determines the contribution rate of the PR index to the near-term future value, was set at 0.67 according to *Halpern et al. (2012)*, but an appropriate value that emphasizes the link between environmental factors and service scores should be considered.

The indicators of each service were limited to environmental investigation indices that have been generally implemented in Japan. For this reason, from a scientific point of view, some of these may be inappropriate indices or models. For example, use of the carbon storage rate as an index of carbon storage is adequate if it is to evaluate the mitigation effect of global warming, but it is limited in that it is based on the existing carbon storage amount as an indicator. In addition, because the number of species used in the evaluation of diversity depends largely on the number of survey sites, a correction model must be implemented for the number of survey sites to eliminate any bias. However, some insufficiency in these types of indicators cannot be avoided in any model owing to data constraints, lack of scientific knowledge, and other similar factors. Recognizing the insufficiency of data and models, nevertheless, we should still work to improve them to the extent possible (*Lowndes et al., 2015*).

## Range of error in scores

Trend scores were evaluated based on the slope of the trajectory of the present status scores over the most recent 5 years. As such, differences in slope estimations are reflected in likely near-term future status and service scores. The 95% confidence intervals (CI) for likely near-term future status obtained by performing linear regression increase the further one moves away from the data used to make the estimate. This occurs because the slopes and intercepts of estimated regression lines have associated variances that are compounded over time. For this reason, we constructed 95% CIs for trend scores based on the standard errors of the slopes of the regression lines. Multiplying this value by

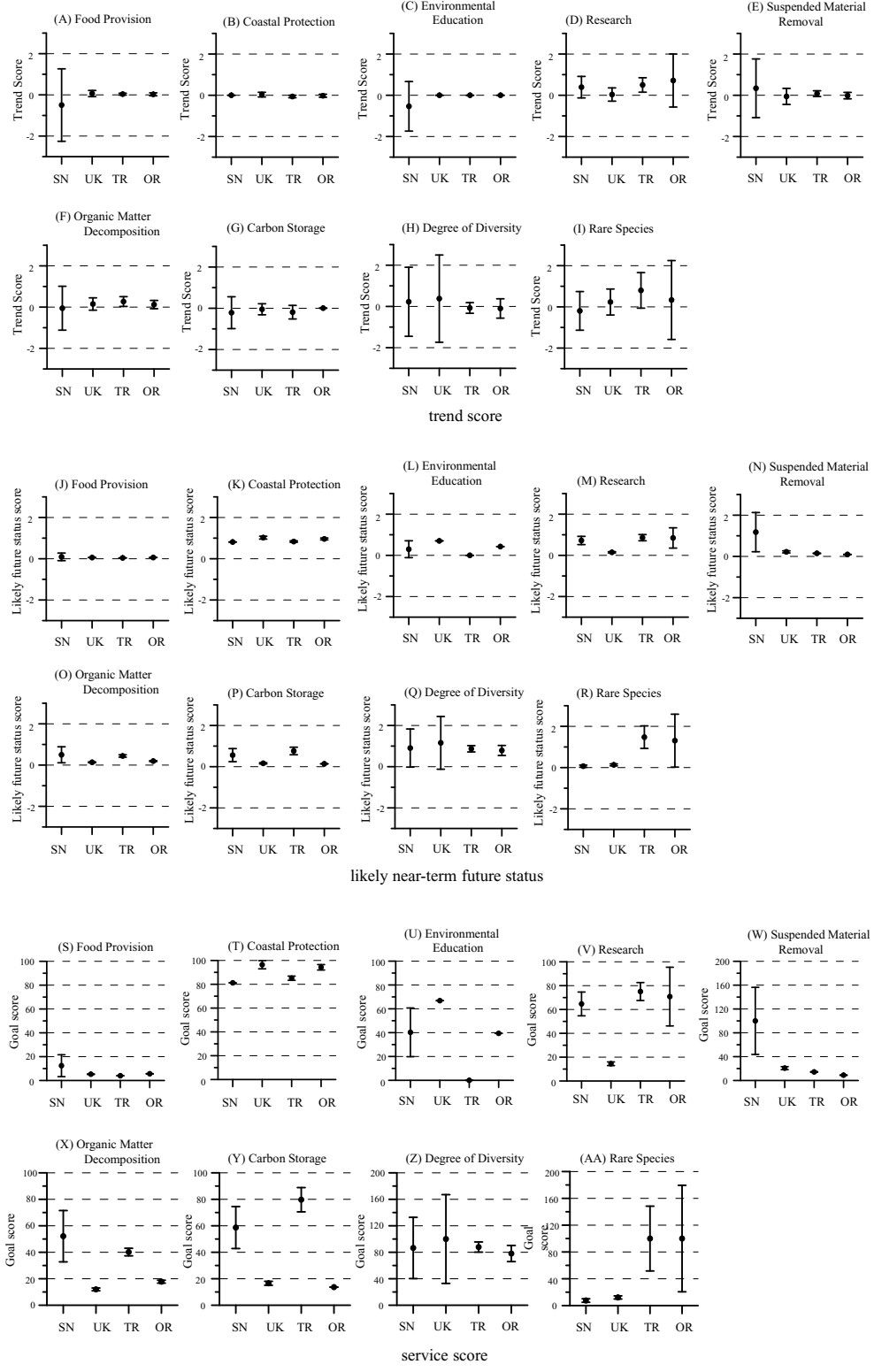

**Figure 5 Ranges of error in scores.** (A–I) trend scores, (J–R) likely near-term future status, and (S–AA) service scores for food provision, coastal protection, environmental education, research, suspended material removal, organic matter decomposition, carbon storage, degree of diversity, and rare species.

5 (representing the years) yields the 95% CIs for the trend scores. The 95% CI based on standard error can easily be estimated by multiplying the standard error by 3 (three degrees of freedom, 95% confidence limits for a two-tailed $t$-distribution). The following equations were used:

$$T_i = 5 \cdot t_i \tag{5}$$

$$T_{i-U} = 5 \cdot T_i + 5 \cdot 3se_i \tag{6}$$

$$T_{i-L} = 5 \cdot T_i - 5 \cdot 3se_i \tag{7}$$

where $T_i$ is the trend score, $T_{i\text{-}U}$ is the upper limit of the 95% CI, $T_{i\text{-}L}$ is the lower limit of the 95% CI, $t_i$ is the slope of the regression line, and $se_i$ is the standard error of the slope.

We estimated CIs for likely near-term future status and service scores for food provision, coastal protection, environmental education, research, suspended material removal, organic matter decomposition, carbon storage, degree of diversity, and rare species for which likely near-term future status were calculated based on past data (Fig. 5). The other services (recreation, historical designation as special sites, and places for everyday rest and relaxation) were not included because there was no past data and trends were not estimated.

For food provision, suspended material removal, organic matter decomposition, and degree of diversity, whose service scores depend on biomass, the error was large in SN. It has been less than 10 years since SN was constructed, and biomass in the area may still be in transition. In addition, SN is located in a port with poor water quality, and environmental impacts such as anoxic water and blue tide often occur and the habitat environment is unstable. In contrast, the error for rare species was greater for the natural tidal flats. This occurred because, in the natural tidal flats, the annual differences in the number of rare species observed was large, whereas in the artificial tidal flats, few rare species were observed in all years so the data were stable from year to year.

Although there were services with a deviation of about ±20, the average deviation was less than ±10. Some of the variation is likely the result of annual changes reflected in the survey data.

## CONCLUSIONS

In this study, we developed a method that can properly quantify services and identify environmental factors to improve services by fully considering the ecosystem of the surrounding waters of four artificial and natural tidal flats. To appropriately reflect the ecological conditions of the surrounding waters in the evaluation results of an environmental improvement project, it is important to create a conceptual model that includes the characteristics of the surrounding waters. The developed method is capable of creating a set of scores for the environmental factors that affect each service. Therefore, it is possible to better understand priority factors for improving the area and the environmental factors for which countermeasures would be effective, all of which can be utilized for more efficient management. However, there are several problems

at this stage in the method development, as noted in the discussion. In future work, we hope to address these problems and create an even more practical evaluation method.

### Funding
This work was supported in part by Grants-in-Aid (KAKENHI, no. 18H04156) and the Environment Research and Technology Development Fund (S-14) of the Ministry of the Environment, Japan. There was no additional external funding received for this study. The funders had no role in study design, data collection and analysis, decision to publish, or preparation of the manuscript.

### Grant Disclosures
The following grant information was disclosed by the authors:
KAKENHI: 18H04156.
Environment Research and Technology Development Fund (S-14) of the Ministry of the Environment, Japan.

### Competing Interests
Tomohiro Kuwae is an Academic Editor for PeerJ. Yugo Mito, Toshiyuki Takahashi and Takanori Sugano are employed by Fukken Co., Ltd.

### Author Contributions
- Tomonari Okada conceived and designed the experiments, performed the experiments, prepared figures and/or tables, authored or reviewed drafts of the paper, approved the final draft.
- Yugo Mito performed the experiments, analyzed the data, contributed reagents/materials/analysis tools, prepared figures and/or tables, authored or reviewed drafts of the paper, approved the final draft.
- Erina Iseri performed the experiments, authored or reviewed drafts of the paper.
- Toshiyuki Takahashi performed the experiments, analyzed the data, contributed reagents/materials/analysis tools, prepared figures and/or tables.
- Takanori Sugano performed the experiments, analyzed the data, contributed reagents/materials/analysis tools, prepared figures and/or tables.
- Yoshihiro B. Akiyama analyzed the data.
- Kenta Watanabe analyzed the data.
- Toko Tanaya analyzed the data.
- Hiroaki Sugino analyzed the data.
- Kanae Tokunaga analyzed the data.
- Takahiro Kubo analyzed the data.
- Tomohiro Kuwae conceived and designed the experiments, authored or reviewed drafts of the paper.

## Data Availability

Raw data are provided in the Supplemental Files.

## Supplemental Information

Supplemental information for this article can be found online at http://dx.doi.org/10.7717/peerj.6234#supplemental-information.

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
