# Peer review of "Method for the quantitative evaluation of ecosystem services in coastal regions"

_PeerJ, doi:10.7717/peerj.6234_

## Round 0.1 · original submission · Major Revisions

· Academic Editor

Major Revisions

Thanks for your submission. I agree with the comments from both reviewers requesting major revisions. More detail on this index is needed, including pros/cons, similarities/differences to OHI, and a model for each sub-index.

Reviewer 1 ·

Basic reporting

English was mostly very good – have indicated a few spelling errors in the attached PDF.

The manuscript develops the indicator/assessment literature relatively well. However, the manuscript would greatly benefit by better introducing tidal flats as these are the subject of the paper. Explaining their characteristics and the difference between man-made and natural tidal flats would help those who are not experts with this ecosystem. The Shiosai Nagisa tidal flat pictured in Figure 4, looks more like a concrete structure than what I would think of as an artificial tidal flat. Explanation of the characteristics is needed up front.

The structure of the article works generally and figures are of sufficient resolution. However, there is a significant portion of data that is not presented and hinders adequate peer review. For instance, from the 12 sub-goals of the index the authors propose, only food provision is presented in terms of the data used, and how scores were calculated. This can be presented in a more refined way such as a table or tables, but equations and descriptions should also really be provided. These can be placed in supplementary materials, but should be available.

Experimental design

The research question could be defined more clearly – it is not clear whether the purpose of the index is to compare between man-made flats and natural flats or just assess all different flats.

While some methods are very well developed, others are vague. It would benefit hugely by more detail on each sub-goal, how the scores were aggregated and how they were weighted.

The authors refer to the theory behind setting reference points but then apply the same reference point (highest score across all areas) to every sub-goal. The reference point will be the main driver behind the index and therefore I would recommend development of a clearer methodology for choosing reference point for each goal – about what should be an ‘optimum’ reference point. The conceptual model approach that the authors’ used for investigating resilience and pressures is an excellent way to think about these ideas.

Validity of the findings

It is difficult to truly understand the validity of the results given there is a lack of information presented for each of the goal models. It is necessary, either in the text or in the supplementary, to provide a model for each sub-index. It is not sufficient to present just one sub-index (food-provision).

Food Provision – reference point is highest catch across all years and all sites. This is not necessarily a desirable reference point – this could be a year subject to intense overfishing. Reference point should be related more to sustainability and can differ between sites.

Water Front Use – Scoring SN as 0 because it is a closed facility is potentially unfair. Might be better to leave this out of the water front use assessment?

Discussion would benefit by being less fractured and a more holistic view of the results. What are the most interesting points? In order to validate the index, results should not simply be accepted. Validating indices is extremely difficult but a more nuanced discussion can help to interpret results better; also, composite indicators can always be iteratively improved over time by refining different areas or utilising better data. Some discussion of this would be welcomed (see ‘Navigating uncertainty in environmental composite indicators’). Interpreting your results and things that look odd – i.e. some scores are 0 for some tidal flats and 100 for others – will likely want refinement. As this type of indicator is overly sensitive (all or nothing) and could have a large impact on the overall index.

Given this it is not entirely clear how the index can be used for management. This ties back to the goal of the index and what it is to be used for.

Comments for the author

Although this paper is described as being modeled on the Ocean Health Index - it is significantly different. The Ocean Health Index theoretically assesses sustainability of large ocean areas, with multiple habitats and a large range of benefits. Adapting the OHI for this scale is a large change and therefore requires justification for many of the decisions taken. The OHI has strong theoretical backing for many of its goals and therefore by trying to adapt the OHI, it means having to justify clearly all your sub-indices (see other regional OHI papers). It may be wiser to pull-back on the comparisons to the Ocean Health Index given how different this is.

Reviewer 2 ·

Basic reporting

Overall, the paper is well written and figures and tables are well presented and support the article. In particular, the introduction Is very well written, thoroughly explained, and flows together quite nicely. There are times where the text can be redundant and a little lengthy which, while I understand the authors want the text to be thorough, can be quite distracting for the reader. I have included some suggestions below for the introduction as well as an annotated pdf, which is attached. Some areas of the methodology are unclear and lack appropriate references and this should certainly be addressed.

Suggested chnages to grammar are below for the Introduction with added comments for the remainder of the paper in the annotated pdf....

81 Halpern et al. (2012) proposed the Ocean Health Index (OHI) as a method for
82 comprehensively quantitating ocean health. The OHI defines healthy oceans as those that
83 sustainably deliver a range of benefits to people now and in the future and enables the
84 comprehensive evaluation of not only benefits gained from ecosystems but also social and
85 economic benefits (Halpern et al., 2014). The OHI scores the state of each ocean against its
86 reference point (Samhouri et al., 2012) which enables the
87 degree of achievement of each goal to be evaluated. The OHI also enables sustainability and
88 changes in policy or regulatory systems to be taken into account and can be adapted flexibly to
89 match the quantity and quality of data, as well as changes in geographic scale
90 (Elfes et al., 2014; Halpern et al., 2015; Lowndes et al. 2015; Seling et al., 2015). One key
91 advantage of the OHI over other approaches is that it enables the scoring and quantification of
92 services that are difficult to quantify in economic terms.

93, 94 Remove this sentence since it doesn’t quite flow with the rest of the paragraph…..”Meanwhile, the biodiversity of coastal areas has been severely reduced due to the habitat degradation of numerous organisms resulting from various human activities.”
96: Remove “As such”
99 Remove “In addition”
103: In
104: addition, since the effect of the environmental improvement project strongly depends on the
105 ecosystem states, it is necessary to carefully consider the states for appropriate
106 evaluation of the project (Zhao et al., 2016). Moreover, because the ecosystem states after
107 creating artificial tidal flat (or wetland) change over time (Palmer, Ambrose & Poff, 1997;
108 Kuwae, 2005), the effect of the environmental improvement project should be evaluated
for several years following completion. Therefore, development of an evaluation method that considers the states and changes of ecosystems is also a challenge in evaluating the benefits of the environmental improvement project.
.

Experimental design

The research question seems to be well defined and relevant to environmental issues outlined in the introduction. While many of the goal and sub-goal criteria are well presented, some are not well referenced or explained at all (please see comments of thresholds for DO, research papers, rare species ranking criteria, etc in annotated pdf). In particular, there was very little information on water front use for research, use for education, use for recreation, and weighting of rare species. How were these sampled, what was the sample size, etc? It seems like none of this is reported in the paper. This makes it difficult to access how rigorous or robust the methodology actually is and the lack of detail needs to be addressed before the paper is considered publishable. I have considerable concerns on the sample design, results, and interpretability of this study if the sample design is not presented.

Another major concern of mine is the impact of not standardizing the wet weight by unit area or unit effort (number of fishing days) for each tidal site. How does area affect other metrics as well considering the two artificial tidal areas are the smallest in area and have the least amount of coastline.

On a positive note, the example the authors provide for calculating scores is nicely written and helped clarify the process of calculating the index for the goals and sub-goals that were well defined.

Validity of the findings

I feel like this paper is missing a major synthesis of the index that is being proposed. What are the strengths of this index? What are the weaknesses? How did area and length of coastline affect metrics, was standardization important? The discussion and conclusion need to be expanded in order to bring the paper back to the original focus of the paper as outlined through the introduction.

Comments for the author

I think the article is interesting but is lacking some key information regarding how the authors calculated their goal and sub goal criteria. Please see my comments regarding the experimental design. The discussion and conclusion sections need additional detail as well in order to give support to the original research question. While the introduction was well written, the conclusion doesn't quite address the original research question and simply focuses on the findings of each site rather than also synthesizing the pros, cons, and caveats of using this index.

---

## Round 0.2 · Minor Revisions

· Academic Editor

Minor Revisions

Thanks for your submission. I agree with the reviewer that minor revisions are necessary to improve the overall flow of the article. Restructuring the sections will improve overall readability. We look forward to receiving your revisions.

Reviewer 1 ·

Basic reporting

The authors have developed the overall paper very well since the previous revision. Development of the introduction has clarified many aspects of the study and the supplementary materials have been vastly improved to give much needed background to the paper.

I feel that the structure of the paper could be reworked to make it easier to follow and generally improve the reading of the paper:

Section 2.7 :Evaluated tidal flats and evaluation year. Bring these descriptions of the tidal flats further up in the paper - either describe them briefly in the introduction or right at the start of the methods.

Lines 132-155: Move these further down the methods. First talk about the overall construction of the index and bring in the sub-indices afterwards.

Section 2.5: This mixes together overall index construction and specific sub-goal examples and that is confusing. Pick one example and keep it concise.

Section 2.6: Move directly below where you talk about likely future state. You move on to pressures and resilience then back to talking about aspects related to LFS.

I think the above restructuring will vastly help the flow of this section, which is crucial in setting up the paper.

Experimental design

I think the experimental design has been improved and the authors have provided adequate justification for their methods.

Validity of the findings

The authors have significantly expanded their discussion to help validate their findings as well as provide discussion and quantification of some uncertainties. They also recognise where improvements might be made.

---

## Round 0.3 · accepted · Accept

· Academic Editor

Accept

Thank you for taking the time to carefully address comments from reviewers.

#